# Gradient Methods for Submodular Maximization

**Hamed Hassani**
ESE Department
University of Pennsylvania
Philadelphia, PA
hassani@seas.upenn.edu

**Mahdi Soltanolkotabi**
EE Department
University of Southern California
Los Angeles, CA
soltanol@usc.edu

**Amin Karbasi**
ECE Department
Yale University
New Haven, CT
amin.karbasi@yale.edu

## Abstract

In this paper, we study the problem of maximizing continuous submodular functions that naturally arise in many learning applications such as those involving utility functions in active learning and sensing, matrix approximations and network inference. Despite the apparent lack of convexity in such functions, we prove that stochastic projected gradient methods can provide strong approximation guarantees for maximizing continuous submodular functions with convex constraints. More specifically, we prove that for monotone continuous DR-submodular functions, all fixed points of projected gradient ascent provide a factor $1/2$ approximation to the global maxima. We also study stochastic gradient methods and show that after $\mathcal{O}(1/\epsilon^2)$ iterations these methods reach solutions which achieve in expectation objective values exceeding $(\frac{\text{OPT}}{2} - \epsilon)$. An immediate application of our results is to maximize submodular functions that are defined stochastically, i.e. the submodular function is defined as an expectation over a family of submodular functions with an unknown distribution. We will show how stochastic gradient methods are naturally well-suited for this setting, leading to a factor $1/2$ approximation when the function is monotone. In particular, it allows us to approximately maximize discrete, monotone submodular optimization problems via projected gradient ascent on a continuous relaxation, directly connecting the discrete and continuous domains. Finally, experiments on real data demonstrate that our projected gradient methods consistently achieve the best utility compared to other continuous baselines while remaining competitive in terms of computational effort.

## 1 Introduction

Submodular set functions exhibit a natural diminishing returns property, resembling concave functions in continuous domains. At the same time, they can be minimized exactly in polynomial time (while can only be maximized approximately), which makes them similar to convex functions. They have found numerous applications in machine learning, including viral marketing [1], dictionary learning [2] network monitoring [3, 4], sensor placement [5], product recommendation [6, 7], document and corpus summarization [8] data summarization [9], crowd teaching [10, 11], and probabilistic models [12, 13]. However, submodularity is in general a property that goes beyond set functions and can be defined for continuous functions. In this paper, we consider the following *stochastic* continuous submodular optimization problem

$$\max_{\boldsymbol{x} \in \mathcal{K}} F(\boldsymbol{x}) \doteq \mathbb{E}_{\theta \sim \mathcal{D}}[F_\theta(\boldsymbol{x})], \tag{1.1}$$

where $\mathcal{K}$ is a bounded convex body, $\mathcal{D}$ is generally an *unknown* distribution, and $F_\theta$'s are continuous submodular functions for every $\boldsymbol{\theta} \in \mathcal{D}$. We also use OPT $\doteq \max_{\boldsymbol{x} \in \mathcal{K}} F(\boldsymbol{x})$ to denote the optimum value. We note that the function $F(\boldsymbol{x})$ is itself also continuous submodular, as a non-negative combination of submodular functions are still submodular [14]. The formulation covers popular

instances of submodular optimization. For instance, when $\mathcal{D}$ puts all the probability mass on a single function, (1.1) reduces to *deterministic* continuous submodular optimization. Another common objective is the *finite-sum* continuous submodular optimization where $\mathcal{D}$ is uniformly distributed over $m$ instances, i.e., $F(x) \doteq \frac{1}{m} \sum_{\theta=1}^{m} F_\theta(x)$.

A natural approach to solving problems of the form (1.1) is to use projected stochastic methods. As we shall see in Section 5, these local search heuristics are surprisingly effective. However, the reasons for this empirical success is completely unclear. The main challenge is that maximizing $F$ corresponds to a nonconvex optimization problem (as the function $F$ is not concave), and a priori it is not clear why gradient methods should yield a reliable solution. This leads us to the main challenge of this paper

> Do projected gradient methods lead to *provably good solutions* for continuous submodular maximization with general convex constraints?

We answer the above question in the affirmative, proving that projected gradient methods produce a competitive solution with respect to the optimum. More specifically, given a general bounded convex body $\mathcal{K}$ and a continuous function $F$ that is monotone, smooth, and (weakly) DR-submodular we show that

- All stationary points of a DR-submodular function $F$ over $\mathcal{K}$ provide a $1/2$ approximation to the global maximum. Thus, projected gradient methods with sufficiently small step sizes (a.k.a. gradient flows) always lead to a solutions with $1/2$ approximation guarantees.

- Projected gradient ascent after $O\left(\frac{L_2}{\epsilon}\right)$ iterations produces a solution with objective value larger than $(\text{OPT}/2 - \epsilon)$. When calculating the gradient is difficult but an unbiased estimate can be easily obtained, the stochastic projected gradient ascent in $O\left(\frac{L_2}{\epsilon} + \frac{\sigma^2}{\epsilon^2}\right)$ iterations finds a solution with objective value exceeding $(\text{OPT}/2 - \epsilon)$. Here, $L_2$ is the smoothness of the continuous submodular function measured in the $\ell_2$-norm, $\sigma^2$ is the variance of the stochastic gradient with respect to the true gradient and OPT is the function value at the global optimum.

- More generally, for weakly continuous DR-submodular functions with parameter $\gamma$ (define in (2.6)) we prove the above results with $\gamma^2/(1+\gamma^2)$ approximation guarantee.

Our result have some important implications. First, they show that projected gradient methods are an efficient way of maximizing the multilinear extension of (weakly) submodular set functions for any submodularity ratio $\gamma$ (note that $\gamma = 1$ corresponds to submodular functions) [2]. Second, in contrast to conditional gradient methods for submodular maximization that require the initial point to be the origin [15, 16], projected gradient methods can start from any initial point in the constraint set $\mathcal{K}$ and still produce a competitive solution. Third, such conditional gradient methods, when applied to the stochastic setting (with a fixed batch size), perform poorly and can produce arbitrarily bad solutions when applied to continuous submodular functions (see [17, Appendix B] in the long version of this paper for an example and further discussion on why conditional gradient methods do not easily admit stochastic variants). In contrast, stochastic projected gradient methods are stable by design and provide a solution with an approximation ratio of at least $1/2$ in expectation. Finally, our work provides a unifying approach for solving the *stochastic submodular maximization problem* [18]

$$f(S) \doteq \mathbb{E}_{\theta \sim \mathcal{D}}[f_\theta(S)], \tag{1.2}$$

where the functions $f_\theta : 2^V \to \mathbb{R}_+$ are submodular set functions defined over the ground set $V$. Such objective functions naturally arise in many data summarization applications [19] and have been recently introduced and studied in [18]. Since $\mathcal{D}$ is unknown, problem (1.2) cannot be directly solved. Instead, [18] showed that in the case of coverage functions, it is possible to efficiently maximize $f$ by lifting the problem to the continuous domain and using stochastic gradient methods on a continuous relaxation to reach a solution that is within a factor $(1 - 1/e)$ of the optimum. In contrast, our work provides a general recipe with $1/2$ approximation guarantee for problem (1.2) in which $f_\theta$'s can be any monotone submodular function.

## 2  Continuous Submodular Maximization

A set function $f : 2^V \to \mathbb{R}_+$, defined on the ground set $V$, is called submodular if for all subsets $A, B \subseteq V$, we have
$$f(A) + f(B) \geq f(A \cap B) + f(A \cup B).$$
Even though submodularity is mostly considered on discrete domains, the notion can be naturally extended to arbitrary lattices [20]. To this aim, let us consider a subset of $\mathbb{R}_+^n$ of the form $\mathcal{X} = \prod_{i=1}^n \mathcal{X}_i$ where each $\mathcal{X}_i$ is a compact subset of $\mathbb{R}_+$. A function $F : \mathcal{X} \to \mathbb{R}_+$ is *submodular* [21] if for all $(\boldsymbol{x}, \boldsymbol{y}) \in \mathcal{X} \times \mathcal{X}$, we have
$$F(\boldsymbol{x}) + F(\boldsymbol{y}) \geq F(\boldsymbol{x} \vee \boldsymbol{y}) + F(x \wedge \boldsymbol{y}), \tag{2.1}$$
where $\boldsymbol{x} \vee \boldsymbol{y} \doteq \max(\boldsymbol{x}, \boldsymbol{y})$ (component-wise) and $\boldsymbol{x} \wedge \boldsymbol{y} \doteq \min(\boldsymbol{x}, \boldsymbol{y})$ (component-wise). A submodular function is monotone if for any $\boldsymbol{x}, \boldsymbol{y} \in \mathcal{X}$ obeying $\boldsymbol{x} \leq \boldsymbol{y}$, we have $F(\boldsymbol{x}) \leq F(\boldsymbol{y})$ (here, by $\boldsymbol{x} \leq \boldsymbol{y}$ we mean that every element of $\boldsymbol{x}$ is less than that of $\boldsymbol{y}$). Like set functions, we can define submodularity in an equivalent way, reminiscent of diminishing returns, as follows [14]: the function $F$ is submodular if for any $\boldsymbol{x} \in \mathcal{X}$, any two distinct basis vectors $\mathrm{e}_i, \mathrm{e}_j \in \mathbb{R}^n$, and any two non-negative real numbers $z_i, z_j \in \mathbb{R}_+$ obeying $\boldsymbol{x}_i + z_i \in \mathcal{X}_i$ and $\boldsymbol{x}_j + z_j \in \mathcal{X}_j$ we have
$$F(\boldsymbol{x} + z_i \mathrm{e}_i) + F(\boldsymbol{x} + z_j \mathrm{e}_j) \geq F(\boldsymbol{x}) + F(\boldsymbol{x} + z_i \mathrm{e}_i + z_j \mathrm{e}_j). \tag{2.2}$$
Clearly, the above definition includes submodularity over a set (by restricting $\mathcal{X}_i$'s to $\{0, 1\}$) or over an integer lattice (by restricting $\mathcal{X}_i$'s to $\mathbb{Z}_+$) as special cases. However, in the remainder of this paper we consider *continuous* submodular functions defined on product of sub-intervals of $\mathbb{R}_+$. We note that when twice differentiable, $F$ is submodular if and only if all cross-second-derivatives are non-positive [14], i.e.,
$$\forall i \neq j, \forall \boldsymbol{x} \in \mathcal{X}, \;\; \frac{\partial^2 F(\boldsymbol{x})}{\partial x_i \partial x_j} \leq 0. \tag{2.3}$$
The above expression makes it clear that continuous submodular functions are not convex nor concave in general as concavity (convexity) implies that $\nabla^2 F \preceq 0$ (resp. $\nabla^2 F \succeq 0$). Indeed, we can have functions that are both submodular and convex/concave. For instance, for a concave function $g$ and non-negative weights $\lambda_i \geq 0$, the function $F(\boldsymbol{x}) = g(\sum_{i=1}^n \lambda_i x_i)$ is submodular and concave. Trivially, affine functions are submodular, concave, and convex. A proper subclass of submodular functions are called *DR-submodular* [16, 22] if for any $\boldsymbol{x}, \boldsymbol{y} \in \mathcal{X}$ obeying $\boldsymbol{x} \leq \boldsymbol{y}$, any standard basis vector $\mathrm{e}_i \in \mathbb{R}^n$, and any non-negative number $z \in \mathbb{R}_+$ obeying $z \mathrm{e}_i + \boldsymbol{x} \in \mathcal{X}$ and $z \mathrm{e}_i + \boldsymbol{y} \in \mathcal{X}$, we have
$$F(z \mathrm{e}_i + \boldsymbol{x}) - F(\boldsymbol{x}) \geq F(z \mathrm{e}_i + \boldsymbol{y}) - F(\boldsymbol{y}). \tag{2.4}$$
One can easily verify that for a differentiable DR-submodular functions the gradient is an antitone mapping, i.e., for all $\boldsymbol{x}, \boldsymbol{y} \in \mathcal{X}$ such that $\boldsymbol{x} \leq \boldsymbol{y}$ we have $\nabla F(\boldsymbol{x}) \geq \nabla F(\boldsymbol{y})$ [16]. When twice differentiable, DR-submodularity is equivalent to
$$\forall i \mathrel{\&} j, \forall \boldsymbol{x} \in \mathcal{X}, \;\; \frac{\partial^2 F(\boldsymbol{x})}{\partial x_i \partial x_j} \leq 0. \tag{2.5}$$
The above twice differentiable functions are sometimes called *smooth* submodular functions in the literature [23]. However, in this paper, we say a differentiable submodular function $F$ is *L-smooth* w.r.t a norm $\| \cdot \|$ (and its dual norm $\| \cdot \|_*$) if for all $\boldsymbol{x}, \boldsymbol{y} \in \mathcal{X}$ we have
$$\| \nabla F(\boldsymbol{x}) - \nabla F(\boldsymbol{x}) \|_* \leq L \| \boldsymbol{x} - \boldsymbol{y} \|.$$
Here, $\| \cdot \|_*$ is the *dual norm* of $\| \cdot \|$ defined as $\| \boldsymbol{g} \|_* = \sup_{\boldsymbol{x} \in \mathbb{R}^n : \| \boldsymbol{x} \| \leq 1} \boldsymbol{g}^T \boldsymbol{x}$. When the function is smooth w.r.t the $\ell_2$-norm we use $L_2$ (note that the $\ell_2$ norm is self-dual). We say that a function is *weakly DR-submodular* with parameter $\gamma$ if
$$\gamma = \inf_{\substack{x, y \in \mathcal{X} \\ \boldsymbol{x} \leq \boldsymbol{y}}} \inf_{i \in [n]} \frac{[\nabla F(\boldsymbol{x})]_i}{[\nabla F(\boldsymbol{y})]_i}. \tag{2.6}$$
See [24] for related definitions. Clearly, for a differentiable DR-submodular function we have $\gamma = 1$. An important example of a DR-submodular function is the multilinear extension [15] $F : [0, 1]^n \to \mathbb{R}$ of a discrete submodular function $f$, namely,
$$F(\boldsymbol{x}) = \sum_{S \subseteq V} \prod_{i \in S} x_i \prod_{j \notin S} (1 - x_j) f(S).$$

We note that for set functions, DR-submodularity (i.e., Eq. 2.4) and submodularity (i.e., Eq. 2.1) are equivalent. However, this is not true for the general submodular functions defined on integer lattices or product of sub-intervals [16, 22].

The focus of this paper is on continuous submodular maximization defined in Problem (1.1). More specifically, we assume that $\mathcal{K} \subset \mathcal{X}$ is a a general bounded convex set (not necessarily down-closed as considered in [16]) with diameter $R$. Moreover, we consider $F_\theta$'s to be monotone (weakly) DR-submodular functions with parameter $\gamma$.

# 3   Background and Related Work

Submodular set functions [25, 20] originated in combinatorial optimization and operations research, but they have recently attracted significant interest in machine learning. Even though they are usually considered over discrete domains, their optimization is inherently related to continuous optimization methods. In particular, Lovasz [26] showed that the Lovasz extension is convex if and only if the corresponding set function is submodular. Moreover, minimizing a submodular set-function is equivalent to minimizing the Lovasz extension.[1] This idea has been recently extended to minimization of strict continuous submodular functions (i.e., cross-order derivatives in (2.3) are strictly negative) [14]. Similarly, approximate submodular maximization is linked to a different continuous extension known as multilinear extension [28]. Multilinear extension (which is an example of DR-submodular functions studied in this paper) is not concave nor convex in general. However, a variant of conditional gradient method, called *continuous greedy*, can be used to approximately maximize them. Recently, Chekuri et al [23] proposed an interesting multiplicative weight update algorithm that achieves $(1 - 1/e - \epsilon)$ approximation guarantee after $\tilde{O}(n^2/\epsilon^2)$ steps for twice differentiable monotone DR-submodular functions (they are also called smooth submodular functions) subject to a polytope constraint. Similarly, Bian et al [16] proved that a conditional gradient method, similar to the continuous greedy algorithm, achieves $(1 - 1/e - \epsilon)$ approximation guarantee after $O(L_2/\epsilon)$ iterations for maximizing a monotone DR-submodular functions subject to special convex constraints called *down-closed* convex bodies. A few remarks are in order. First, the proposed conditional gradient methods cannot handle the general stochastic setting we consider in Problem (1.1) (in fact, projection is the key). Second, there is no near-optimality guarantee if conditional gradient methods do not start from the origin. More precisely, for the continuous greedy algorithm it is necessary to start from the $\mathbf{0}$ vector (to be able to remain in the convex constraint set at each iteration). Furthermore, the $\mathbf{0}$ vector must be a feasible point of the constraint set. Otherwise, the iterates of the algorithm may fall out of the convex constraint set leading to an infeasible final solution. Third, due to the starting point requirement, they can only handle special convex constraints, called down-closed. And finally, the dependency on $L_2$ is very suboptimal as it can be as large as the dimension $n$ (e.g., for the multilinear extensions of some submodular set functions, see [17, Appendix B] in the long version of this paper). Our work resolves all of these issues by showing that projected gradient methods can also approximately maximize monotone DR-submodular functions subject to general convex constraints, albeit, with a lower $1/2$ approximation guarantee.

Generalization of submodular set functions has lately received a lot of attention. For instance, a line of recent work considered DR-submodular function maximization over an integer lattice [29, 30, 22]. Interestingly, Ene and Nguyen [31] provided an efficient reduction from an integer-lattice DR-submodular to a submodular set function, thus suggesting a simple way to solve integer-lattice DR-submodular maximization. Note that such reductions cannot be applied to the optimization problem (1.1) as expressing general convex body constraints may require solving a continuous optimization problem.

# 4   Algorithms and Main Results

In this section we discuss our algorithms together with the corresponding theoretical guarantees. In what follows, we assume that $F$ is a weakly DR-submodular function with parameter $\gamma$.

## 4.1 Characterizing the quality of stationary points

We begin with the definition of a stationary point.

**Definition 4.1** *A vector $\boldsymbol{x} \in \mathcal{K}$ is called a stationary point of a function $F : \mathcal{X} \to \mathbb{R}_+$ over the set $\mathcal{K} \subset \mathcal{X}$ if $\max_{\boldsymbol{y} \in \mathcal{K}} \langle \nabla F(x), \boldsymbol{y} - \boldsymbol{x} \rangle \leq 0$.*

Stationary points are of interest because they characterize the fixed points of the Gradient Ascent (GA) method. Furthermore, (projected) gradient ascent with a sufficiently small step size is known to converge to a stationary point for smooth functions [32]. To gain some intuition regarding this connection, let us consider the GA procedure. Roughly speaking, at any iteration $t$ of the GA procedure, the value of $F$ increases (to the first order) by $\langle \nabla F(\boldsymbol{x}_t), \boldsymbol{x}_{t+1} - \boldsymbol{x}_t \rangle$. Hence, the progress at time $t$ is at most $\max_{\boldsymbol{y} \in \mathcal{K}} \langle \nabla F(\boldsymbol{x}_t), \boldsymbol{y} - \boldsymbol{x}_t \rangle$. If at any time $t$ we have $\max_{\boldsymbol{y} \in \mathcal{K}} \langle \nabla F(\boldsymbol{x}_t), \boldsymbol{y} - \boldsymbol{x}_t \rangle \leq 0$, then the GA procedure will not make any progress and it will be stuck once it falls into a stationary point.

The next natural question is how small can the value of $F$ be at a stationary point compared to the global maximum? The following lemma relates the value of $F$ at a stationary point to OPT.

**Theorem 4.2** *Let $F : \mathcal{X} \to \mathbb{R}_+$ be monotone and weakly DR-submodular with parameter $\gamma$ and assume $\mathcal{K} \subseteq \mathcal{X}$ is a convex set. Then,*

  *(i) If $\boldsymbol{x}$ is a stationary point of $F$ in $\mathcal{K}$, then $F(\boldsymbol{x}) \geq \frac{\gamma^2}{1+\gamma^2} \text{OPT}$.*

  *(ii) Furthermore, if $F$ is L-smooth, gradient ascent with a step size smaller than $1/L$ will converge to a stationary point.*

The theorem above guarantees that all fixed points of the GA method yield a solution whose function value is at least $\frac{\gamma^2}{1+\gamma^2} \text{OPT}$. Thus, all fixed point of GA provide a factor $\frac{\gamma^2}{1+\gamma^2}$ approximation ratio. The particular case of $\gamma = 1$, i.e., when $F$ is DR-submodular, asserts that at any stationary point $F$ is at least $\text{OPT}/2$. This lower bound is in fact tight. In the long version of this paper (in particular [17, Appendix A]) we provide a simple instance of a differentiable DR-Submodular function that attains $\text{OPT}/2$ at a stationary point that is also a *local maximum*.

We would like to note that our result on the quality of stationary points (i.e., first part of Theorem 4.2 above) can be viewed as a simple extension of the results in [33]. In particular, the special case of $\gamma = 1$ follows directly from [28, Lemma 3.2]. See the long version of this paper [17, Section 7] for how this lemma is used in our proofs. However, we note that the main focus of this paper is whether such a stationary point can be found efficiently using stochastic schemes that do not require exact evaluations of gradients. This is the subject of the next section.

## 4.2 (Stochastic) gradient methods

We now discuss our first algorithmic approach. For simplicity we focus our exposition on the DR submodular case, i.e., $\gamma = 1$, and discuss how this extends to the more general case in the long version of this paper ([17, Section 7]). A simple approach to maximizing DR submodular functions is to use the (projected) Gradient Ascent (GA) method. Starting from an initial estimate $\boldsymbol{x}_1 \in \mathcal{K}$ obeying the constraints, GA iteratively applies the following update

$$\boldsymbol{x}_{t+1} = \mathcal{P}_{\mathcal{K}} \left( \boldsymbol{x}_t + \mu_t \nabla F(\boldsymbol{x}_t) \right). \tag{4.1}$$

Here, $\mu_t$ is the learning rate and $\mathcal{P}_{\mathcal{K}}(\boldsymbol{v})$ denotes the Euclidean projection of $\boldsymbol{v}$ onto the set $\mathcal{K}$. However, in many problems of practical interest we do not have direct access to the gradient of $F$. In these cases it is natural to use a stochastic estimate of the gradient in lieu of the actual gradient. This leads to the Stochastic Gradient Method (SGM). Starting from an initial estimate $\boldsymbol{x}_0 \in \mathcal{K}$ obeying the constraints, SGM iteratively applies the following updates

$$\boldsymbol{x}_{t+1} = \mathcal{P}_{\mathcal{K}} \left( \boldsymbol{x}_t + \mu_t \boldsymbol{g}_t \right). \tag{4.2}$$

Specifically, at every iteration $t$, the current iterate $\boldsymbol{x}_t$ is updated by adding $\mu_t \boldsymbol{g}_t$, where $\boldsymbol{g}_t$ is an unbiased estimate of the gradient $\nabla F(\boldsymbol{x}_t)$ and $\mu_t$ is the learning rate. The result is then projected onto the set $\mathcal{K}$. We note that when $\boldsymbol{g}_t = \nabla F(\boldsymbol{x}_t)$, i.e., when there is no randomness in the updates, then

---

**Algorithm 1** (Stochastic) Gradient Method for Maximizing $F(x)$ over a convex set $\mathcal{K}$

---

**Parameters:** Integer $T > 0$ and scalars $\eta_t > 0$, $t \in [T]$
**Initialize:** $\boldsymbol{x}_1 \in \mathcal{K}$
**for** $t = 1$ **to** $T$ **do**
    $\boldsymbol{y}_{t+1} \leftarrow \boldsymbol{x}_t + \eta_t \boldsymbol{g}_t$,
       where $\boldsymbol{g}_t$ is a random vector s.t. $\mathbb{E}[\boldsymbol{g}_t | \boldsymbol{x}_t] = \nabla F(\boldsymbol{x}_t)$
    $\boldsymbol{x}_{t+1} = \arg\min_{\boldsymbol{x} \in \mathcal{K}} \|\boldsymbol{x} - \boldsymbol{y}_{t+1}\|_2$
**end for**
Pick $\tau$ uniformly at random from $\{1, 2, \ldots, T\}$.
**Output** $\boldsymbol{x}_\tau$

---

the SGM updates (4.2) reduce to the GA updates (4.1). We detail the SGM method in Algorithm 1. As we shall see in our experiments detained in Section 5, the SGM method is surprisingly effective for maximizing monotone DR-submodular functions. However, the reasons for this empirical success was previously unclear. The main challenge is that maximizing $F$ corresponds to a nonconvex optimization problem (as the function $F$ is not concave), and a priori it is not clear why gradient methods should yield a competitive ratio. Thus, studying gradient methods for such nonconvex problems poses new challenges:

> Do (stochastic) gradient methods converge to a stationary point?

The next theorem addresses some of these challenges. To be able to state this theorem let us recall the standard definition of smoothness. We say that a continuously differentiable function $F$ is $L$-smooth (in Euclidean norm) if the gradient $\nabla F$ is $L$-Lipschitz, that is $\|\nabla F(\boldsymbol{x}) - \nabla F(\boldsymbol{y})\|_{\ell_2} \le L \|\boldsymbol{x} - \boldsymbol{y}\|_{\ell_2}$. We also defined the diameter (in Euclidean norm) as $R^2 = \sup_{\boldsymbol{x},\boldsymbol{y} \in \mathcal{K}} \frac{1}{2} \|\boldsymbol{x} - \boldsymbol{y}\|_{\ell_2}^2$. We now have all the elements in place to state our first theorem.

**Theorem 4.3 (Stochastic Gradient Method)** *Let us assume that $F$ is $L$-smooth w.r.t. the Euclidean norm $\|\cdot\|_{\ell_2}$, monotone and DR-submodular. Furthermore, assume that we have access to a stochastic oracle $\boldsymbol{g}_t$ obeying*

$$\mathbb{E}[\boldsymbol{g}_t] = \nabla F(\boldsymbol{x}_t) \quad and \quad \mathbb{E}\left[ \|\boldsymbol{g}_t - \nabla F(\boldsymbol{x}_t)\|_{\ell_2}^2 \right] \le \sigma^2.$$

*We run stochastic gradient updates of the form (4.2) with $\mu_t = \frac{1}{L + \frac{\sigma}{R}\sqrt{t}}$. Let $\tau$ be a random variable taking values in $\{1, 2, \ldots, T\}$ with equal probability. Then,*

$$\mathbb{E}[F(\boldsymbol{x}_\tau)] \ge \frac{\text{OPT}}{2} - \left( \frac{R^2 L + OPT}{2T} + \frac{R\sigma}{\sqrt{T}} \right). \tag{4.3}$$

**Remark 4.4** *We would like to note that if we pick $\tau$ to be a random variable taking values in $\{2, \ldots, T-1\}$ with probability $\frac{1}{(T-1)}$ and $1$ and $T$ each with probability $\frac{1}{2(T-1)}$ then*

$$\mathbb{E}[F(\boldsymbol{x}_\tau)] \ge \frac{\text{OPT}}{2} - \left( \frac{R^2 L}{2T} + \frac{R\sigma}{\sqrt{T}} \right).$$

The above results roughly state that $T = \mathcal{O}\left( \frac{R^2 L}{\epsilon} + \frac{R^2 \sigma^2}{\epsilon^2} \right)$ iterations of the stochastic gradient method from any initial point, yields a solution whose objective value is at least $\frac{\text{OPT}}{2} - \epsilon$. Stated differently, $T = \mathcal{O}\left( \frac{R^2 L}{\epsilon} + \frac{R^2 \sigma^2}{\epsilon^2} \right)$ iterations of the stochastic gradient method provides in expectation a value that exceeds $\frac{\text{OPT}}{2} - \epsilon$ approximation ratio for DR-submodular maximization. As explained in Section 4.1, it is not possible to go beyond the factor $1/2$ approximation ratio using gradient ascent from an arbitrary initialization.

An important aspect of the above result is that it only requires an unbiased estimate of the gradient. This flexibility is crucial for many DR-submodular maximization problems (see, (1.1)) as in many cases calculating the function $F$ and its derivative is not feasible. However, it is possible to provide a good un-biased estimator for these quantities.

We would like to point out that our results are similar in nature to known results about stochastic methods for convex optimization. Indeed, this result interpolates between the $\frac{1}{\sqrt{T}}$ for stochastic smooth optimization, and the $1/T$ for deterministic smooth optimization. The special case of $\sigma = 0$ which corresponds to Gradient Ascent deserves particular attention. In this case, and under the assumptions of Theorem 4.3, it is possible to show that $F(\boldsymbol{x}_T) \geq \frac{\text{OPT}}{2} - \frac{R^2 L}{T}$, without the need for a randomized choice of $\tau \in [T]$.

Finally, we would like to note that while the first term in (4.3) decreases as $1/T$, the pre-factor $L$ could be rather large in many applications. For instance, this quantity may depend on the dimension of the input $n$ (see [17, Appendix C] in the extended version of this paper). Thus, the number of iterations for reaching a desirable accuracy may be very large. Such a large computational load causes (stochastic) gradient methods infeasible in some application domains. It is possible to overcome this deficiency by using stochastic mirror methods (see [17, Section 4.3] in the extended version of this paper).

## 5 Experiments

In our experiments, we consider a movie recommendation application [19] consisting of $N$ users and $n$ movies. Each user $i$ has a user-specific utility function $f_i$ for evaluating sets of movies. The goal is to find a set of $k$ movies such that in expectation over users' preferences it provides the highest utility, i.e., $\max_{|S| \leq k} f(S)$, where $f(S) \doteq \mathbb{E}_{i \sim \mathcal{D}}[f_i(S)]$. This is an instance of the stochastic submodular maximization problem defined in (1.2). We consider a setting that consists of $N$ users and consider the empirical objective function $\frac{1}{N} \sum_{j=1}^{N} f_i$. In other words, the distribution $\mathcal{D}$ is assumed to be uniform on the integers between 1 and $N$. We can then run the (discrete) greedy algorithm on the empirical objective function to find a good set of size $k$. However, as $N$ is a large number, the greedy algorithm will require a high computational complexity. Another way of solving this problem is to evaluate the multilinear extension $F_i$ of any sampled function $f_i$ and solve the problem in the continuous domain as follows. Let $F(\boldsymbol{x}) = \mathbb{E}_{i \sim \mathcal{D}}[F_i(\boldsymbol{x})]$ for $x \in [0,1]^n$ and define the constraint set $\mathcal{P}_k = \{\boldsymbol{x} \in [0,1]^m : \sum_{i=1}^{n} x_i \leq k\}$. The discrete and continuous optimization formulations lead to the same optimal value [15]:

$$\max_{S:|S| \leq k} f(S) = \max_{\boldsymbol{x} \in \mathcal{P}_k} F(\boldsymbol{x}).$$

Therefore, by running the stochastic versions of projected gradient methods, we can find a solution in the continuous domain that is at least $1/2$ approximation to the optimal value. By rounding that fractional solution (for instance via randomized Pipage rounding [15]) we obtain a set whose utility is at least $1/2$ of the optimum solution set of size $k$. We note that randomized Pipage rounding does not need access to the value of $f$. We also remark that projection onto $\mathcal{P}_k$ can be done very efficiently in $O(n)$ time (see [18, 34, 35]). Therefore, such an approach easily scales to big data scenarios where the size of the data set (e.g. number of users) or the number of items $n$ (e.g. number of movies) are very large.

In our experiments, we consider the following baselines:

  (i) Stochastic Gradient Ascent (SG): We use the step size $\mu_t = c/\sqrt{t}$ and mini-batch size $B$. The details for computing an unbiased estimator for the gradient of $F$ are given in [17, Appendix D] of the extended version of this paper.

 (ii) Frank-Wolfe (FW) variant of [16]: We use $T$ to denote the total number of iterations and $B$ to denote mini-batch sizes (we further let $\alpha = 1, \delta = 0$, see Algorithm 1 in [16] for more details).

(iii) Batch-mode Greedy (Greedy): We run the vanilla greedy algorithm (in the discrete domain) in the following way. At each round of the algorithm (for selecting a new element), $B$ random users are picked and the function $f$ is estimated by the average over the $B$ selected users.

To run the experiments we use the MovieLens data set. It consists of 1 million ratings (from 1 to 5) by $N = 6041$ users for $n = 4000$ movies. Let $r_{i,j}$ denote the rating of user $i$ for movie $j$ (if such a rating does not exist we assign $r_{i,j}$ to 0). In our experiments, we consider two well motivated objective functions. The first one is called "facility location" where the valuation function by user $i$ is defined

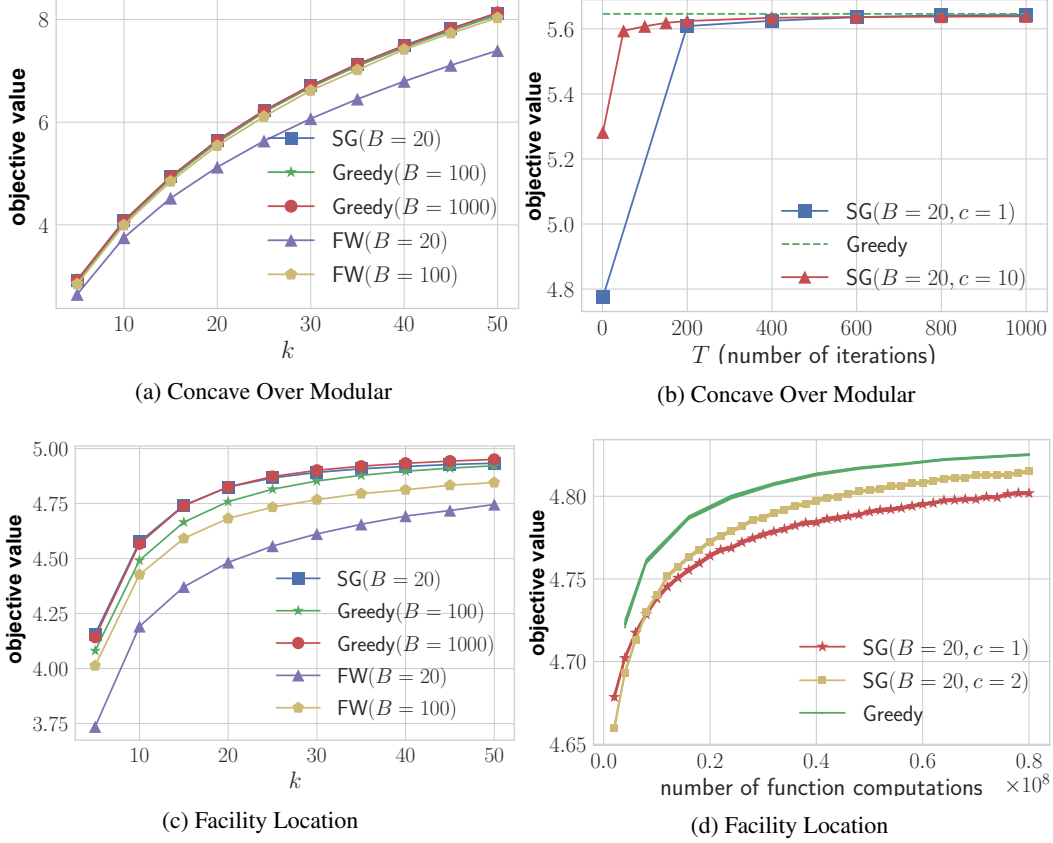

(a) Concave Over Modular

(b) Concave Over Modular

(c) Facility Location

(d) Facility Location

Figure 1: (a) shows the performance of the algorithms w.r.t. the cardinality constraint $k$ for the concave over modular objective. Each of the continuous algorithms (i.e., SG and FW) run for $T = 2000$ iterations. (b) shows the performance of the SG algorithm versus the number of iterations for fixed $k = 20$ for the concave over modular objective. The green dashed line indicates the value obtained by Greedy (with $B = 1000$). Recall that the step size of SG is $c/\sqrt{t}$. (c) shows the performance of the algorithms w.r.t. the cardinality constraint $k$ for the facility location objective function. Each of the continuous algorithms (SG and FW) run for $T = 2000$ iterations. (d) shows the performance of different algorithms versus the number of simple function computations (i.e. the number of $f_i$'s evaluated during the algorithm) for the facility location objective function. For the greedy algorithm, larger number of function computations corresponds to a larger batch size. For SG larger time corresponds to larger iterations.

as $f(S, i) = \max_{j \in S} r_{i,j}$. In words, the way user $i$ evaluates a set $S$ is by picking the highest rated movie in $S$. Thus, the objective function is equal to

$$f_{\text{fac}}(S) = \frac{1}{N} \sum_{i=1}^{N} \max_{j \in S} r_{i,j}.$$

In our second experiment, we consider a different user-specific valuation function which is a concave function composed with a modular function, i.e., $f(S, i) = (\sum_{j \in S} r_{i,j})^{1/2}$. Again, by considering the uniform distribution over the set of users, we obtain

$$f_{\text{con}}(S) = \frac{1}{N} \sum_{i=1}^{N} \Big( \sum_{j \in S} r_{i,j} \Big)^{1/2}.$$

Note that the multilinear extensions of $f_1$ and $f_2$ are neither concave nor convex.

Figure 1 depicts the performance of different algorithms for the two proposed objective functions. As Figures 1a and 1c show, the FW algorithm needs a much higher mini-batch size to be comparable

in performance to our stochastic gradient methods. Note that a smaller batch size leads to less computational effort (using the same value for $B$ and $T$, the computational complexity of FW and SGA is almost the same). Figure 1b shows that after a few hundred iterations SG with $B = 20$ obtains almost the same utility as the Greedy method with a large batch size ($B = 1000$). Finally, Figure 1d shows the performance of the algorithms with respect to the number of times the single functions ($f_i$'s) are evaluated. This further shows that gradient based methods have comparable complexity w.r.t. the Greedy algorithm in the discrete domain.

## 6  Conclusion

In this paper we studied gradient methods for submodular maximization. Despite the lack of convexity of the objective function we demonstrated that local search heuristics are effective at finding approximately optimal solutions. In particular, we showed that all fixed point of projected gradient ascent provide a factor $1/2$ approximation to the global maxima. We also demonstrated that stochastic gradient and mirror methods achieve an objective value of $\mathrm{OPT}/2 - \epsilon$, in $\mathcal{O}(\frac{1}{\epsilon^2})$ iterations. We further demonstrated the effectiveness of our methods with experiments on real data.

While in this paper we have focused on convex constraints, our framework may allow non-convex constraints as well. For instance it may be possible to combine our framework with recent results in [36, 37, 38] to deal with general nonconvex constraints. Furthermore, in some cases projection onto the constraint set may be computationally intensive or even intractable but calculating an approximate projection may be possible with significantly less effort. One of the advantages of gradient descent-based proofs is that they continue to work even when some perturbations are introduced in the updates. Therefore, we believe that our framework can deal with approximate projections and we hope to pursue this in future work.

**Acknowledgments**

This work was done while the authors were visiting the Simon's Institute for the Theory of Computing. A. K. is supported by DARPA YFA D16AP00046. The authors would like to thank Jeff Bilmes, Volkan Cevher, Chandra Chekuri, Maryam Fazel, Stefanie Jegelka, Mohammad-Reza Karimi, Andreas Krause, Mario Lucic, and Andrea Montanari for helpful discussions.

## Footnotes

[1]The idea of using stochastic methods for submodular minimization has recently been used in [27].

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
