[Reviews · NeurIPS 2017]

Reviewer 1



This paper study the problem of continuous submodular maximization subject to a convex constraint. The authors show that for monotone weakly DR-submodular functions, all the stationary points of this problem are actually good enough approximations (gamma^2 / (1 + gamma^2), where gamma is the DR factor) of the optimal value. They also show that stochastic projected gradient descent and mirror descent converge to a stationary point in O(1/eps^2) iterations. The paper is clear and well presented, with few typos. The results presented can handle more general constraints than prior work and can address the stochastic setting, but at the cost of a worst approximation factor (1/2 for DR-submodular functions as opposed to 1 - 1/e). Some comments/questions: -It is worth mentioning that stochastic projected gradient descent was also used recently for submodular minimization in "Chakrabarty, D., Lee, Y. T., Sidford, A., & Wong, S. C. W. Subquadratic submodular function minimization. STOC 2017." - For the deficiency example of FW in A.2, it would be clearer to present the example with a minibatch setting, with m_i,n = 1/(b+1) batch size, the same example should follow through, but it would make it clearer that a slightly larger minibatch size won't fix the problem. - Would this deficiency example hold also for the algorithm "Measured continuous greedy" in "Feldman, Moran, Joseph Naor, and Roy Schwartz. "A unified continuous greedy algorithm for submodular maximization." FOCS, 2011, whose updates are slightly different from FW? - You reference the work [20] on p.2, but I did not find this work online? - Can you add what are other examples of L-smooth continuous submodular functions, other than the multilinear extension? - For a more convincing numerical comparison, it would be interesting to see the time comparison (CPU time not just iterations) against FW. - What is the y-axis in the figures? Some typos: -line 453: f_ij - f_i - f_j (flipped) -lemma B.1: right term in the inner product should be proj(y) - x. -lemma B.2: first term after equality should be D_phi(x,y) - Eq after line 514 and line 515: grad phi(x+) (nabla missing) - In proof of lemma B.5: general norm instead of ell_2 - Eq after line 527 and 528: x_t and x^* flipped in inner product.

Reviewer 2



Summary: The paper proves a 1/2-approximation guarantee for fixed points of monotone continuous DR-submodular functions. It also proves guarantees for stochastic gradient and mirror methods. Overall, I think the theoretical results in the paper are interesting. However, the presentation of the paper is not good and confusing, and it reduces the quality of the whole paper. Some specific comments: 1. The paper contains so many typos and grammar mistakes that are really distracting. 2. From line 94, the definition says that the cross-second-derivatives are non-negative, but in Equation (2.3) they are <= 0. 3. It is not clear to me what the L_2 mentioned on line 109 is? And what does that sentence mean? 4. In Theorem 4.2, the function should be weakly DR-submodular. 5. The definition of the projection in Definition 4.5 is confusing. I think this sentence needs to be rewritten. 6. Theorems 4.2, 4.3, and 4.6 seem to prove results for different types of functions. For instance, Theorem 4.2 is for weakly DR-submodular functions, Theorem 4.3 is only for DR-submodular functions, and Theorem 4.6 is for submodular functions. The paper should make them clear and discuss this more. 7. For Theorem 4.6, does the function F need to be monotone? If not, why? 8. In the experiment, since FW was proposed for the non-stochastic case, the paper should also compare the methods in this setting (i.e. using full gradients instead of mini-batches). 9. Since the main contributions of the paper are the theoretical results, it would be useful to include some proof sketches of the theorems. The paper can save more space by, for example, reducing the repeated description of the mirror ascent algorithm in Theorem 4.6.

Reviewer 3



This paper considers algorithms for the maximization of continuous (weakly) submodular functions in the stochastic setting over a bounded convex set, and proposes using projected SGD. SGD and projected SGD methods have been widely studied for convex stochastic problems, but the problem considered here is not convex. These methods have also been studied recently for nonconvex L-smooth stochastic optimization problems where rates of convergence to stationary points are obtained. In my view, what is novel in this paper is to combine this existing results with a continuous DR-submodular assumption, which allows the authors to show that all stationary points have values of at least OPT/2, so they can get a 1/2 approximation guarantee (Theorem 4.2). This is a nice approach that brings continuous optimization ideas to a field that is often studied from a discrete viewpoint, which I think is timely and relevant. My vote is a "clear accept", but I feel addressing the comments/questions below can indeed make it a "strong accept": 1. I suggest the authors give a brief proof outline of teh main theorems in the main text; I did look at the proofs in the supplement, but it is better to have a sense of what part of proofs follow from similar proofs of convergnce of nonconvex SGD and what parts are specific to using the submodularity structure. 2. I enjoyed reading the paper, and I like the approach taken. However I was left with the question of why the continuous greedy algorithm can't be adapted to the stochastic setting. What is the bottleneck here? Also, in comparison to FW methods, starting at zero is cited as a weakness, but this seems not to be a problem in most examples. 3. Equation 2.6 is wrong as written; as it does not make sense to divide by a vector. (easy to fix, but I surprised at the sloppiness here given that the paper well written overall). 4. Just for clarity, in eq 1.1, state clearly that F_\theta(x) is submodular in x for every \theta. 5. Can some nonconvex constraint sets which have an easy projection be handled as well? 6. What if the projection onto set K can be computed only approximately?